# OpenReview forum: "CPMöbius: Iterative Coach–Player Reasoning for Data-Free Reinforcement Learning"
_ICLR.cc/2026/Conference — Submitted to ICLR 2026_

### Official Review · Reviewer_cag2 · 2025-10-25

**Soundness:** 1
**Presentation:** 2
**Contribution:** 1
**Rating:** 2
**Confidence:** 3

**Summary:**

This paper focuses on improving the reasoning capability of LLMs without extra data by a sollaborative coach-player paradigm. The coach LLM proposes instruction to maximize the player's accuracy on validation set. Then the player LLM  conduct self-training on these generated instructions with pseudo-labels. Experiments on Qwen2.5-Math and OctoThinker are conducted to validate the proposed method.

**Strengths:**

- The proposed method is simple and straightforward.
- Most parts of this manuscript are easy to follow up with.

**Weaknesses:**

- Limited novelty and contribution. There have been several works focuses on leveraging data-free or label-free RL for reasoning. From my reading of this paper, the proposed method seems very similar to absolute zero. However, direct comparison or indirect but comprehensive discussion are missing. Besides, the author should at least mention EMPO[1] and Intuitor[2] in the related works. I also enourage to include TTRL or EMPO baselines if it is possible.
- The experiments results are noticeably lower than state-of-art results when considering the same Base model. For example, on Qwen2.5-Math-7B, zero RL can achieve 40% and 65% accuracy on AIME24 and AMC23 benchmarks respectively [4]. Even without groundtruth, unsupervised RL can achieve 20% and 65% accuracy on these two datasets [3]. Although the proposed method is devised in a data-free manner, such a large margin raise concerns about the effectiveness of the proposed coach-player training paradigm. Besides, it is also quesntionable whether the performance gain of the proposed method is higher than prompt engineering (especially for Base model).
- The so-called OOD setting does not make sense to me. In the experiments, the model is trained and evaluated on both Math domain. Besides, all the Base model are pretrained with math-specific data corpus. Thus the OOD generation capability of the proposed method is overclaimed.
- Serveral words and sentences are weired to me and seldom ocurred in acedamic writting. I doubt if this manuscript is written by LLMs without enough polishment.
- Figure 1 and 2 are not informative. They are not helpful for the reviewer to follow up with the core idea.

[1] Right Question is Already Half the Answer: Fully Unsupervised LLM Reasoning Incentivization, NeurIPS25

[2] Learning to Reason without External Rewards

[3] DeepMath-103K: A Large-Scale, Challenging, Decontaminated, and Verifiable Mathematical Dataset for Advancing Reasoning

**Questions:**

- Why there are only 100 steps in Figure 5 but not 1000?

---

> ### Author Response · Authors · 2025-11-25
>
> Thank you for your detailed review。Below, we respond to your concerns and clarify key aspects of our work.
>
> ---
>
> **W1**:
> First, you mentioned that "the proposed method seems very similar to Absolute Zero." For Absolute Zero's proposer-solver mechanism, it's a self-play between individual models, emphasizing self-evolution through adversarial competition between the proposer and solver. Absolute Zero heavily relies on the code executor as a verifiable environment to provide absolutely objective feedback, including whether the proposer's problem statement is complete and whether the solver provides the correct answer (i.e., verifying the deterministic reward). This is a very hard constraint. Therefore, Absolute Zero is a framework that relies on the code executor to provide "verifiable rewards" for reinforcement learning training. For CPMöbius's coach-player mechanism, the emphasis is on co-evolution between the coach and player, focusing on collaboration to improve the player model's performance ceiling. CPMöbius doesn't require a hard verifier to provide absolutely objective feedback and validate deterministic rewards. The core driving force of the coach is to determine whether the player has improved. The main role of the AMC validation set is to provide a positive or negative force for the coach to judge whether the player has improved. This is similar to how, in the real world, coaches provide different training strategies based on each athlete's level, and after training, athletes participate in a competition. The coach adjusts the next training strategy based on changes in different metrics in the competition results. The coach's main goal is to help athletes reach a new level, while also improving the coach's teaching skills. The figure below illustrates the main differences between different works. Therefore, we chose R-zero as one of our baselines. R-zero is a very high-quality work, and its self-play mechanism does not rely on verifiable supervision from a hard verifier, making it more suitable as our baseline. However, when using R-zero to train OpenMath-Nemotron-1.5B, we found that R-zero's challenger failed to ask appropriate questions, ultimately leading to training failure. Therefore, we believe that models like the coach, which can ask questions, are very important.
>
> Secondly, you mentioned the EMPO method. Similarly, we also tested another Entropy Minimization baseline, RENT[4], a fully unsupervised RL method that requires no external reward or ground-truth answers, and instead uses the model’s entropy of its underlying distribution as an intrinsic reward. Our experimental results also show that we have an improvement compared to the Entropy Minimization method RENT. For TTRL, the player now provides the reward through majority voting and is trained using GRPO, so it is basically equivalent to TTRL. The main motivation of CPMöbius is to improve the upper limit of the player model through the paradigm of coach player, whether it is a model after RL or based on large-scale SFT. (Please understand that due to GPU resource limitations, we can only prioritize DeepMath-Zero-Math-7B based on CPMöbius for training)
> | | externally verifiable supervision | Without externally verifiable supervision |
> | :--- | :--- | :--- |
> | **self-play** | Absolute-zero [1] | R-zero [2] |
> | **co-evlove** | CURE [3] | CPMöbius |
>
> [1] Zhao, Andrew, et al. "Absolute zero: Reinforced self-play reasoning with zero data." arXiv preprint arXiv:2505.03335 (2025).
> [2] Huang, Chengsong, et al. "R-Zero: Self-Evolving Reasoning LLM from Zero Data." arXiv preprint arXiv:2508.05004 (2025).
> [3] Wang, Yinjie, et al. "Co-evolving llm coder and unit tester via reinforcement learning." arXiv preprint arXiv:2506.03136 (2025).
> [4] Prabhudesai, Mihir, et al. "Maximizing Confidence Alone Improves Reasoning." arXiv preprint arXiv:2505.22660 (2025).

---

> > ### Author Response · Authors · 2025-11-25
> >
> > **W2**:
> > We agree that it is important to put our numbers in the right context. Our work aims to take the first step toward a data-free reinforcement learning (RL) paradigm in which the model is improved entirely through a coach-player collaborative loop without any external training data.
> > The works cited in [3] rely on substantial external reasoning corpora (DeepMath-103K and other curated competition-style datasets) and often also perform RL directly on distributions highly overlapped with the evaluation benchmarks. In contrast, CPMöbius explicitly restricts the training setting to be data-free: 1) we do not use DeepMath-103K, MATH-hard, or any AIME/AMC training split during RL; 2) all training problems are generated online by the Coach; 3) AMC23 is only used as held-out evaluation.
> >
> > As a result, our method answers a different question: “Can we obtain non-trivial improvements in mathematical reasoning purely from verifiable rewards on coach-generated tasks, without relying on any additional training corpus?” We believe this The data-free setting is orthogonal and complementary to zero-RL or DeepMath-style setups rather than directly comparable in absolute numbers. We reproduced DeepMath-Zero-Math-7B, trained it on CPMöbius, and evaluated it using the test script provided by DeepMath-103K on the mathematical reasoning benchmark in the table below. We adopted pass@1 accuracy (averaged over 16 samples) as the metric, and fixed the decoding parameters to temperature=0.6, top p=0.95, and max tokens=32K. There was an improvement of +1.3 in OOD Average and +1.2 in Average. This demonstrates that the current Coach setting can still guide the highly capable DeepMath-Zero-Math-7B with performance improvements.
> >
> > | | AMC | AIME2024 | AIME2025 | Minerva | Math | Olympiad | OOD avg | avg |
> > | :--- | :--- | :--- | :--- | :--- | :--- | :--- | :--- | :--- |
> > | **DeepMath-Zero-Math-7B**
> > | **Paper Report** | 74.7 | 34.2 | 23.5 | 49.5 | 86.9 | 52.3 | 49.3 | 53.5 |
> > | **Replicated Report** | 73.6 | 33.6 | 21.2 | 49.1 | 86.6 | 51.8 | 48.4 | 52.7 |
> > | **CPMöbius** | **74.7** | **33.8** | **23.5** | **50.0** | **88.2** | **52.9** | **49.7** | **53.9** |
> >
> > ---
> >
> > **W3**:
> > In the caption of **Table 1** in the original paper, we mentioned the definition of OOD Average. **"OOD Average denotes the out-of-distribution performance, calculated as the mean across all benchmarks excluding AMC datasets, since RENT was trained on AMC and CPMöbius utilized AMC as validation. This separation ensures fair comparison by distinguishing in-distribution (AMC) from out-of-distribution generalization performance. Bold values ​​indicate best performance for each metric."** Since the entropy minimization method Rent uses AMC as training data, for fairness, we use the mathematical inference dataset, which was not used as training data, as the OOD performance.
> >
> > ---
> >
> > **W4**:
> > I apologize for any inconvenience caused by certain issues with the academic writing. Could you please specify which "Serveral" elements were involved? I can then correct them in the text. Furthermore, I assure you that this manuscript was not written using LLMs; we only used LLMs to polish a portion of the content.
> >
> > ---
> >
> > **W5**:
> > From our respect, I can explain these two pictures again here. **Figure 1**: CPMöbius starts with the coach proposing tasks of suitable difficulty. The player learns by solving these tasks, then reviews on a predefined validation set. Finally, the coach adjusts the next training plan based on the player’s performance.
> > **Figure 2**: The illustration on the conceptual layered architecture on the design of CPMöbius. The iterative process includes four stages. Coach designs training plan: The coach gives instructions of suitable difficulty based on the player’s current ability. Player trains: The player executes each instruction multiple times, uses majority voting to get pseudo-labels, and updates with GRPO. Player evaluates results: The updated player is tested on a prepared validation set, and the accuracy of validation is recorded. Coach adjusts plan: The coach adjusts with updates REINFORCE, using the player’s performance on both the proposed instructions and the validation set as rewards.
> >
> > ---
> >
> > **Questions**:
> > Because the obvious performance divergence occurs rapidly:
> > The variant "w/o Coach SFT warm-up" (green line) suffers from catastrophic performance degradation (accuracy drops from ~0.36 to ~0.20) within the first 100 steps. Similarly, "w/o Coach Update" (orange line) fails to establish a positive learning trajectory. Only "w/o Instruction Filter" shows an unstable upward trend, but by step 100, the performance with the ablation model scoring approximately 4% lower than CPMöbius. These early failures provide sufficient empirical evidence that removing these components leads to immediate instability. Extending these failed runs to 1000 steps would not change the conclusion. CPMöbius (blue line) establishes a stable upward trend immediately.

---

### Official Review · Reviewer_7o3P · 2025-10-26

**Soundness:** 2
**Presentation:** 3
**Contribution:** 3
**Rating:** 4
**Confidence:** 4

**Summary:**

The paper introduces CPMöbius, a two-agent training loop for improving math reasoning without using external supervised datasets. The two agents are a Coach who generates problems, and a Player who solves them using reinforcement learning.
The Player is updated using GRPO, a critic-free RL method that normalizes rewards, while the Coach receives a reward based on how much the Player’s validation accuracy improves on a fixed benchmark (AMC), making the Coach cooperative rather than adversarial.
The authors report that this process improves math reasoning for multiple model sizes (1.5B–7B) and outperforms baselines such as RENT and R-Zero on both overall and out-of-distribution benchmarks.

**Strengths:**

1. The Coach’s reward is directly linked to the Player’s improvement on a validation set (Δt), providing a clear optimization signal rather than relying on self-assessed progress. This makes the training objective interpretable and well-grounded.
2. The paper outlines a concrete four-stage loop with pseudo-code showing the alternation between question generation, Player updates, validation, and Coach updates. The implementation is more transparent than in many prior self-play RL works.
3. Experiments span models at different training stages, including pretrained, SFT, and RL-tuned. It indicates the method’s generality beyond a single setup. Disabling the Coach update, warm start, or difficulty filter each degrades performance, showing that each component contributes meaningfully to the final results.

**Weaknesses:**

### “Data-free” is oversold

 - The paper's "data-free" and "no external training data" claims seem to overlook significant dependencies on human-curated data.

 - The Coach isn't built from scratch. It's initialized from Qwen2.5-7B-Instruct and then "cold-started" using the PRIME Eurus-2-RL-Data.

 - That dataset consists of human-produced math problems, meaning the Coach inherits a substantial amount of math-specific signal before the main loop even begins.

 - The chosen Player models (like OpenMath-Nemotron-1.5B or Qwen2.5-Math-7B-Instruct) are selected precisely because they have already undergone extensive math SFT or RL tuning on large-scale, human-created math datasets.

 - The "progress reward" isn't self-contained; it relies on the AMC benchmark as a held-out validation set in every single round. AMC is a standard benchmark of human-written problems.

While the incremental fine-tuning data is self-generated, the entire system is scaffolded and guided by existing human data. The claims should be qualified. A more accurate description might be "no new human-curated supervision is introduced after initialization" or that improvement is driven by "self-generated instructions validated against a fixed external benchmark".

---

### “Majority vote = ground truth” can self-confirm errors

The Player generates n answers, then takes majority vote as pseudo-label y*. Every sample that matches y* is rewarded as “correct".
This assumes that the Player’s most common answer is correct. That is not always true, especially early in training. This could reinforce systematic mistakes and drive mode collapse toward confident but wrong heuristics.

The paper partially addresses this by:

 -  limiting training to questions where the Player isn’t trivially consistent (accuracy <0.8) but also not random (accuracy >0.2), and
 -  adding KL regularization to keep the Player near the reference policy.

But we still don’t see a direct analysis of “Are pseudo-labels actually correct?” or “Does the Player ever get stuck in a bad attractor?” The consistency plots (Fig. 4) show difficulty trends, but they don’t measure correctness against ground truth.
This is a safety hole. The method is only as truthful as its self-verifier.

---

### Baseline fairness

They compare against RENT and R-Zero. They say R-Zero “failed” on OpenMath-Nemotron-1.5B because “the challenger could not be trained", implying instability.

That’s interesting, but if R-Zero was never able to produce any challenger for that model, that’s not a clean head-to-head comparison. It’s more like: “Our method trains where theirs exploded". That’s valuable but should be stated clearly as robustness, not purely accuracy. Baseline parity matters a lot because small % gains (2–5 point average) can come from prompt engineering and sampling strategy choices.

If those issues are addressed, this is a solid submission. It has a crisp story (“stop adversarial self-play, make the teacher care about student progress”) and shows consistent quantitative gains across heterogeneous base models, which is rare in this space.

**Questions:**

### Is AMC leaking into training?

The authors evaluate OOD performance separately from AMC to argue generalization. They justify this because AMC is used as validation during training and thus becomes in-distribution.
That’s fair, but practically:
The Coach reward directly uses Δt = Acc_val(new) − Acc_val(old) on AMC.

So AMC is shaping the Coach policy.
In other words, AMC is not just “a metric", it’s part of the RL signal for the Coach. At that point AMC is effectively training data, even if you call it “validation". This weakens the purity of the “out-of-distribution” story. The paper is transparent about splitting “overall avg” vs “OOD avg", which is good, but will likely still push on whether this is just AMC overfitting plus some transfer.

**Follow-up I’d want:**

 -  Show performance on AMC variants that are not exactly the subset used online for Δt during training (e.g., hold out 20% AMC for Coach signal and 80% for final report, or rotate validation sets).
 - Show whether the Coach starts overfitting to AMC style (short algebra questions vs e.g. Olympiad-style constructive proofs). Right now we only see final numbers, not qualitative drift.

---

### Can the authors provide more compute / stability details?

 - How often does the Coach collapse (e.g., starts generating garbage or trivial variants)?
 - How often does Player training diverge (very high KL, mode collapse, etc.)?
 - GPU hours / improvement curve.

You already plot training curves for AMC accuracy and “answer consistency". It would be great to add similar curves for: question length growth, answer length compression, etc., but quantify them more precisely (you mention this qualitatively in Section 4.3 and appendices).

Furthermore, please be explicit about how RENT and R-Zero were run:

 - same base checkpoints?
 - same rollout counts (n=16)?
 - same sampling temperatures / pass@k evaluation protocol?

 ---

### Can the authors be clear about the positioning?

I think the real pitch here is:

> “We turn math reasoning improvement into an online cooperative curriculum design problem, where a Coach LLM learns to propose maximally educational tasks for a Player LLM, and the only scalar of truth is measured generalization gain on a fixed verifiable benchmark".

That is new compared to standard RLHF (needs human labels), standard RLVR (needs verifiers with ground truth), and adversarial self-play (unstable).
I would foreground that as the main conceptual contribution, more than “data-free".

---

> ### Author Response · Authors · 2025-11-26
>
> Thank you for your detailed review。Below, we respond to your concerns and clarify key aspects of our work.
>
> ---
>
> **W1 & Q1** :
> We acknowledge that AMC acts as a meta-level optimization objective because its performance influences the coach's transition signal ($\Delta_t$). However, we don't think this constitutes "data leakage" in tradition, nor does it lead to overfitting. Concerns regarding supervised purity and generalization ability will be discussed below:
> The difference between Supervision and Validation Signal is crucial to clarify that the AMC dataset is never used for gradient updates.
>
> The Player model never sees the instruction, ground truth, or specific error modes of the AMC problems during training.
> The Coach only receives a scalar signal that is an accuracy fluctuation to decide how to update the student. This mechanism is similar to standard validation-based model selection or early stopping, where a validation set only influences the final model choice without being part of the training data backpropagation. Therefore, the model cannot memorize AMC solutions or exist hacking the metric through parameter optimization on AMC samples.
>
> The reviewer points out that if AMC shapes the policy, we must ensure the model won't overfitting to AMC problems (I'm not sure what do you mean about AMC-style, it's kind of like generated by LLMs). Our results on Out-Of-Distribution benchmarks provide the strongest evidence for solving this concern. As shown in our main results table for Qwen2.5-Math-1.5B, the model achieves consistent gains on Olympiad from 23.4 to 26.9 and Minerva from 16.3 to 28.0. The Olympiad dataset consists of complex, constructive proofs, which are structurally distinct from AMC's multiple choice format. If the coach were biasing the model solely towards AMC problems, we would expect performance in the Olympiad to stop or degrade. Instead, we observe a obvious improvement +3.5 .
>
> Due to the current shortage of GPU resources, we will add the experiments you mentioned in subsequent paper updates. However, as I mentioned before, the purpose of this validation set is simply to provide the coach model with a positive or negative force to determine whether the player model needs adjustment to the current plan. Therefore, the other validation sets serve the same purpose. The reason we chose AMC is because its difficulty is moderate compared to other mathematical inference benchmarks, which is more conducive to the coach's control over the design.
>
> ---
>
> **W2** :
> The work of TTRL[1] demonstrates that majority voting, a test-time scaling method, is a very efficient and suitable way to promote RL training and reward shaping. Judging from the improvement results of TTRL and Player Model, this method of using pseudo-labels is effective. Just like a lucky hit does not necessarily reinforce the correct one, but it can suppress the wrong one. In many scenarios beyond human capabilities, where the reward model cannot provide the correct reward, and where there is no verifiable reward, majority voting is a relatively efficient way to solve the reward shaping problem. Therefore, we do not need to pay too much attention to whether pseudo-labels are actually correct or wrong. Based on the experimental results, Player model does not get stuck in a bad attractor.
> [1] Zuo, Yuxin, et al. "Ttrl: Test-time reinforcement learning." arXiv preprint arXiv:2504.16084 (2025).
>
> ---
>
> **W3** :
> Thank you very much for considering this a solid submission. Indeed, our main goal is to assist the coach in continuously improving the player's abilities within their own capabilities, while the player also continuously improves the coach's coaching abilities. To give you a clearer understanding of the issues R-Zero encountered while training Openmath-Nemotron-1.5B, I have provided a more detailed explanation below:
> When training Openmath-Nemotron-1.5B, R-Zero aims to train the model to generate questions wrapped within ```<question></question>``` tags, which are then extracted for solver training. However, during our reproduction, we observed that the trained Openmath-Nemotron-1.5B model frequently failed to comply with the required formatting. Specifically, the model exhibited the following behaviors:
> - It often did not wrap questions in the specified ```<question></question>``` tags. Instead, it used various alternative formats such as starting with "Consider the problem:" or "Problem:", which could not be properly extracted by the R-Zero pipeline.
> - It generated a significant amount of repetitive and meaningless content.
> - It occasionally reproduced the prompt itself verbatim, such as outputting strings like:
> ```python
> <question>\n{The full problem statement on one or more lines}\n</question>\n\n
> ```
> As a result, the solver could not obtain usable training questions, causing the pipeline to fail.

---

> > ### Author Response · Authors · 2025-11-26
> >
> > **Q2** :
> > We are happy to provide the requested details to ensure reproducibility and transparency. We will include a dedicated "Computational Cost and Stability Analysis" section in the appendix of the final revision.
> >
> > I don't quite understand what you mean by "Coach collapse" here. But if you mean generating meaningless instructions, in our experiments, we rarely observed the coach degenerating into generating meaningless instructions. The instruction filter and the Coach reward mechanism structurally prevent this from happening. If the coach generates unsolvable or meaningless problems, the player's performance will decline, resulting in a negative reward. Therefore, the coach is immediately penalized and redirected back to an effective instruction manifold.
> >
> > I was also a little confused about what the Player training divergence meant. However, throughout the training process, we explicitly monitored the KL divergence between the Player model and the reference model to detect signs of instability. We did not observe any training divergence or pattern collapse. We will update the training log in the appendix of the paper. In the training log plot, the KL divergence remained consistently low and bounded. For example, when using the Qwen2.5-Math-1.5B model, the KL loss peaked at approximately 0.01 and tended to stabilize. For the Nemotron model, the KL loss remained very stable at around 0.002. There was no situation where the KL distribution collapsed to an extremely small value and stopped changing; the KL here exhibited normal fluctuations with small gradients, indicating that the model maintained healthy diversity.
> >
> > For all models in the main experiment, we used the same base checkpoint without additional training. For rollout counts and temperature, we used the training parameters reported in the RENT and R-Zero papers to ensure training fairness. Regarding the evaluation protocol, we stated in the paper, "To ensure robustness, we employ benchmark-specific sampling strategies calibrated to each benchmark’s difficulty: mean@32 for AIME benchmarks, mean@10 for AMC, mean@6 for Minerva, mean@5 for MATH-500, and mean@3 for Olympiad-Bench." All evaluation sampling temperatures were based on the model's training temperatures.
> >
> > ---
> >
> > **Q3** :
> > Thank you very much for your insightful comments and suggestions regarding our work. Regarding the "data-free" aspect, we might emphasize that after the CPMöbius framework begins training, no external data is involved. Perhaps "no external supervision" would be more appropriate. We think we also should highlight the co-evolve between the two models through collaboration. We will also elaborate on how this approach differentiates us from existing paradigms: Compared to RLHF: We eliminate the need for human preference labels in the loop. Compared to standard RLVR: We no longer rely on a per-sample ground truth verifiers. Compared to adversarial self-play: We transform the dynamic mechanism from a zero-game that leads to instability to a cooperative game, where the coach is rewarded for the player's progress rather than their failure. In the final version, we plan to position "data-free" as an inherent consequence of the method itself, rather than its definition.

---

> ### Author Response · Authors · 2025-12-02
>
> **Q1**:
> In response to the reviewer's requirement, we also conduct the experiments to show performance on AMC variants that are not exactly the subset used online for Δt during training. In this settings, we hold out 20% AMC for Coach signal and 80% for final report. The results are as follows:
>
> | | AMC (remaining 80%) | AMC | AIME2024 | AIME2025 | Minerva | MATH | Olympiad | OOD avg | avg |
> | :--- | :--- | :--- | :--- | :--- | :--- | :--- | :--- | :--- | :--- |
> | **Qwen2.5-Math-1.5B**
> | Base Model | 38.8 | 34.6 | 6.2 | 2.8 | 16.3 | 56.2 | 23.4 | 19.8 | 23.3 |
> | CPMöbius (20% AMC) | 41.2 | 37.7 | 8.6 | 4.4 | 19.1 | 59.9 | 27.1 | **23.8** | **26.1** |
>
> The results show that CPMöbius achieves superior performance compared to base model when we hold out 20% AMC for Coach signal and 80% for final report. CPMöbius (with 20% AMC for validation) consistently achieves the highest accuracy on remaining 80% AMC and other OOD datasets. Therefore, the results can prove that our method does not take advantage of data leakage and show strong out-of-distribution generalization ability.

---

### Official Review · Reviewer_EwJr · 2025-10-27

**Soundness:** 3
**Presentation:** 3
**Contribution:** 2
**Rating:** 6
**Confidence:** 4

**Summary:**

This paper introduces CPMöbius, a collaborative Coach-Player framework for data-free reinforcement learning aimed at improving mathematical reasoning in large language models. The Coach model generates instructional tasks calibrated to the Player's current capability frontier, while the Player model is trained via GRPO using majority-voted pseudo-labels from self-consistency. The Coach receives rewards based on both the Player's training performance and validation accuracy improvements, creating a cooperative optimization loop. Experimental results show improvements over unsupervised baselines across multiple mathematical reasoning benchmarks.

**Strengths:**

- The shift from adversarial to cooperative multi-agent learning is conceptually interesting and appears more stable than prior adversarial approaches.
- The paper is well written and easy to follow.

**Weaknesses:**

Please see my detailed questions and concerns below.

**Questions:**

- What prevents the Coach from simply memorizing patterns from the validation set rather than learning general curriculum design principles? With only AMC as feedback, how do you ensure the Coach learns transferable instruction generation?
- How do you handle the case when $\Delta t$ is negative? Does the Coach receive negative rewards, and if so, how does this affect REINFORCE gradient estimation stability?
- What proportion of generated instructions are rejected by the filter at different training stages?
- How do you ensure the Coach doesn't collapse into generating trivially easy problems that consistently fall within the 0.2-0.8 range?
- What is the distribution of problem types generated by the Coach over training?
- What proportion of the improvement comes from the cooperative framework versus simply having more compute for self-training? Is this a fair comparison to RENT/R-Zero in terms of computational budget?

---

> ### Author Response · Authors · 2025-11-27
>
> Thank you for your detailed review。Below, we respond to your concerns and clarify key aspects of our work.
>
> ---
> Q1 :
> This is a common question. Let me start by giving an abstract example. Why do I say that "the Coach can only score points by making the Players stronger"? Imagine yourself as a soccer coach. You can't play the game (like not being able to do AMC directly), and you haven't seen who the opponents are (like not having seen specific AMC questions). You can only create training plans (like generating instructions). After the game, you only know the score: win or lose ($\Delta t$). In this situation, to ensure continued wins, the only thing you can do is comprehensively improve the players' physical fitness and skills. You can't teach players "tactics against a specific opponent" because you don't know who the opponent is.
>
> In CPMöbius, the Coach does not train on specific validation problems. Instead, at each outer iteration, it receives a scalar signal summarizing the Player's advantage value for each instruction and a difference of step t and t-1 on the AMC validation set under the current curriculum. The Coach's policy is defined over instructional choices and optimized through policy gradient on this scalar feedback. During training, the coach model does not see the instructions, true values, or specific error patterns of the AMC problem. Therefore, the model cannot memorize AMC solutions, nor can it exploit metrics by optimizing parameters on AMC samples. Our results on Out-Of-Distribution benchmarks provide the strongest evidence for solving this concern. As shown in our main results table for Qwen2.5-Math-1.5B, the model achieves consistent gains on Olympiad from 23.4 to 26.9 and Minerva from 16.3 to 28.0. The Olympiad dataset consists of complex, constructive proofs, which are structurally distinct from AMC's multiple choice format. If the coach were biasing the model solely towards AMC problems, we would expect performance in the Olympiad to stop or degrade. Instead, we observe an obvious improvement +3.5 .
>
> ---
> Q2 :
> We clarify how $\Delta_t$ is defined, how negative values are handled, and what this implies for the stability of the REINFORCE gradient. In our implementation, the Coach reward at iteration $t$ is defined as the difference in Player validation performance:
>
> $$
> R_i^{\text{Coach}} = R_i^{\text{Player}} \cdot \Delta_t
> $$
>
> where $R_i^{\text{Player}} = \frac{1}{n}\sum_{j=1}^n r_{i,j}$ represents the average training reward achieved by the Player on instruction $x_i$, and
>
> $$
> \Delta_t = Acc_{val}(\pi_{\phi_{t+1}}^{\text{P}}) - Acc_{val}(\pi_{\phi_t}^{\text{P}})
> $$
>
>
>
> measures the Player's accuracy improvement on the validation set. This quantity can be positive if the current curriculum improves the Player or negative if the curriculum harms the Player. We feed $R_i^{\text{Coach}}$ directly as the scalar reward using the batch of instruction–reward pairs $\{(x_i, R_i^{\text{Coach}})\}_{i=1}^m$ into the Coach objective:
>
> $$
> \nabla_\theta J(\theta) = \frac{1}{m}\sum_{i=1}^m R_i^{Coach} \nabla_\theta \log \pi_\theta^{C}(x_i).
> $$
>
> which is a standard REINFORCE estimator with a signed reward signal. A negative $R$ reverses the gradient direction. This explicitly penalizes the Coach, decreasing the probability of generating the specific types of instructions that led to the Player's regression. This is a corrective mechanism.
> We acknowledge that using raw REINFORCE with negative rewards can lead to high variance in gradient estimation, which is a known theoretical challenge. In the reported experiments, we computed the gradients using the raw return without a baseline subtraction. Despite the high variance inherent in this approach, we observed that the Coach successfully converged and the entropy of Coach does not collapse. Even if the signal from $R_i^{\text{Player}} \cdot \Delta_t$ is noisy at the step level, it still provided a sufficient directional gradient over the course of training to guide the Coach toward valid curriculum design.
> We agree that adding a baseline to reduce variance is a standard and effective trick in RL. While our method proved relatively robust enough to work without it, implementing a baseline is quite straightforward and would likely further smooth the training dynamics and potentially accelerate convergence in future iterations.
>
> ---
> Q3 :
> For Qwen2.5-Math-1.5B, in experimental observations, approximately 40%-60% were filtered out in the early stages of training, and approximately 10%-20% were filtered out after 400 steps.
>
> ---

---

> > ### Author Response · Authors · 2025-11-27
> >
> > Q4 :
> > Empirically, this is possible. In our experiments, we rarely observed the coach generating meaningless instructions. The instruction filter and coach reward mechanism structurally prevent this from happening. If the coach generates unsolvable or meaningless problems, the player's performance will decline, resulting in a negative reward. Therefore, the coach is immediately penalized and guided back to valid instructions. If the model consistently fails to generate appropriate problems, it will fail to train on OpenMath-Nemotron-1.5B, similar to R-zero's failure.
> >
> > ---
> > Q5 :
> > I will show the 3 instructions for steps of 0, 100, 200, 400
> >
> > Step 0:
> > - A line passing through the point (2,4) intersects the parabola $ y^2=8x $ at exactly one point. How many such lines are there?
> > - Given the quadratic equation $x^{2}-3x-5=0$, find the coefficients of the quadratic term and the constant term after completing the square.
> > - Given sets $A=\\{x|x^2-3x+2=0\\}$ and $B=\\{x|x^2-ax+a-1=0\\}$.\n(1) If $a=3$, find the set $A \\cup B$;\n(2) If $A \\cap B = B$, find the range of values for the real number $a$.
> >
> > Step 100:
> > - A car with a mass of 800 kg is moving at 20 m/s on a horizontal surface. The brakes are applied, and it comes to a stop over a distance of 100 meters. Assuming the braking process can be approximated as uniformly decelerated motion:\n\n1. What is the average force exerted on the car by the road?\n2. What is the work done by the braking force in this process?
> > - Find the smallest positive integer \\( n \\) such that in any 2-coloring of the complete graph on \\( n \\) vertices, there exist two monochromatic triangles that share exactly one common vertex.
> > - In triangle $ABC$, let $\\overline{AB} = 7\\sqrt{2}$, and let the coordinates of points A and B be (0,0) and $(7\\sqrt{2}, 0)$, respectively. Suppose that the internal angle bisector of $\\angle BAC$ intersects line segment $\\overline{BC}$ at point D such that $\\overline{AD} = \\overline{AC}$. Find the length of $\\overline{AC}$.
> >
> > Step 200:
> > - In $\\triangle ABC$, $AC=6$, $\\cos B=\\dfrac{1}{3}$, $C=\\dfrac{\\pi }{4}$.\n\n(I) Find the value of $ \\sin A $;\n\n(II) Find the value of $\\overrightarrow{BC} \\cdot \\overrightarrow{CA}$
> > - A point $ Q $ is randomly selected from the rectangular region with vertices $(0,0), (3,0)$, $(3,2), (0,2)$. What is the probability that $Q$ is closer to the origin than it is to the point $(4,2)$?
> > - A circle has its center at $(0,k)$, with $k > 6$, and is tangent to the lines $y=2x$, $y=-2x$ and $y=6$. What is the radius of this circle?
> >
> > Step 400:
> > - Find the smallest value of \\(z\\) such that there exists an integer \\(n > 17\\) with \\(n^2 \\equiv z \\pmod{50}\\)
> > - Find the equation of the circle that passes through points $A(0,1)$ and $B(1,-2)$, and whose center lies on the line $x+y+5=0$.
> > - A polynomial $p(x)$ is defined as follows: $[ p(x) = \\frac{x^3 - 5x^2 + 8x + 12}{(x+2)(x-3)^2}]$ Express $p(x)$ in the form: $$[p(x) = \\frac{D}{x+2} + \\frac{C}{x-3} + \\frac{B}{(x-3)^2}]$$ Find the product of all possible values of $D$.
> >
> > Step 0 can be categorized as rule-based something that can be mastered through rote memorization. Step 100, involving graph theory, can be categorized as abstract logic, which is a weakness in large models. Steps 200 and 400, involving number theory, can be categorized as deep specialization, the most challenging part.

---

> ### Author Response · Authors · 2025-11-27
>
> ---
> **Q6** :
> The training process of R-Zero involves alternating phases: first training a 5-step questioner, followed by a 15-step solver, repeated three times, resulting in a total of 60 steps, solver global batch size: 128, number of rollouts: 5, challenger global batch size: 128, number of rollouts: 4. Thus, we utilized the checkpoint from our 60th step, which the parameter is: both coach and solver train batch size: 16, both coach and solver number of rollouts: 16, and compared it with R-Zero’s final training outcomes. The context length of all models remains consistent. The results are as follows:
>
> | | AMC | AIME2024 | AIME2025 | Minerva | MATH | Olympiad | OOD avg | avg |
> | :--- | :--- | :--- | :--- | :--- | :--- | :--- | :--- | :--- |
> | **Qwen2.5-Math-1.5B** |
> | R-zero | 39.2 | 9.8 | 5.0 | 19.3 | 62.4 | 26.8 | **24.7** | **27.1** |
> | CPMöbius | 40.0 | 10.3 | 4.9 | 17.3 | 59.3 | 29.3 | 24.2 | 26.9 |
> | **OctoThinker-3B-Hybrid-Zero**|
> | R-zero | 25.9 | 2.0 | 0.3 | 14.6 | 58.1 | 22.3 | 19.5 | 20.5 |
> | CPMöbius | 24.7 | 7.3 | 1.0 | 22.4 | 58.1 | 26.1 | **23.0** | **23.3** |
> | **Qwen2.5-Math-7B-Instruct**
> | R-zero | 50.5 | 9.5 | 7.4 | 32.7 | 83.3 | 38.1 | 34.2 | 36.9 |
> | CPMöbius | 49.0 | 11.3 | 9.6 | 34.9 | 79.5 | 38.2 | **34.7** | **37.1** |
>
> As shown, our method slightly underperforms RZero on the Qwen2.5-Math-1.5B model but achieves stronger performance on the other two models. Moreover, our approach can further enhance model capabilities. According to Section 5.4 of the RZero paper, its performance converges after three iterations. Therefore, we believe that under comparable computational budgets, our method still holds an advantage. And the improvement of model capability stems from our framework raising the performance upper bound, rather than simply leveraging additional compute.
>
> ---

---

### Official Review · Reviewer_zyLm · 2025-11-01

**Soundness:** 3
**Presentation:** 3
**Contribution:** 3
**Rating:** 6
**Confidence:** 3

**Summary:**

The paper proposes a two‑agent cooperative Coach–Player framework to improve LLM mathematical reasoning without external training data for the Player. The Coach generates tasks calibrated to the Player’s current ability; the Player solves them and is trained with GRPO using verifiable rewards obtained via majority vote pseudo‑labels. The paper reports consistent gains over unsupervised baselines (RENT, R‑Zero) across four base models and six math benchmarks, with particularly large improvements on Minerva and MATH.

**Strengths:**

1. The method is novel and goal is clear. The Coach is rewarded by instruction‑level Player reward multiplied by the global validation improvement, directly incentivizing tasks that cause real learning instead of adversarial “gotchas.”
2. The paper shows consistent gains across models and benchmarks. In Table 1, this method improves Qwen2.5 Math 7B by 4.9, and shows large Minerva jumps (e.g., 34.6 to 44.9 for 7B; 16.3 to 28.0 for 1.5B).
3. The ablation study is complete. Removing Coach updates, the difficulty filter, or the Coach warm‑up degrades results (Table 2), and Figs. 3–4 show steady AMC validation gains and adaptive difficulty.

**Weaknesses:**

1. Although Player training is “data‑free,” Sec. 4.1 says the Coach is initialized with Qwen2.5 7B Instruct and "a preliminary cold‑start phase on mathematical problems sourced from PRIME Eurus‑2‑RL‑Data". This undercuts the top‑line "no external training data" message and should be framed as Player data‑free, Coach warmed up with external math data. Please quantify how much PRIME data is used, and show results without any warm‑up (beyond the ablation) across all base models, not just on Qwen2.5‑Math‑1.5B.
2. The Coach reward uses global $\Delta_t$ for all instructions in a batch (Eq. 6). This yields credit assignment ambiguity: one “good” instruction can mask others, and variance could be high. A baseline or control variate for $\Delta_t$, or per‑instruction contribution estimates (e.g., off‑policy influence functions or leave‑one‑out $\Delta_t$ ), would strengthen learning stability and attribution claims. No variance or stability metrics are reported.
3. No confidence intervals or multiple‑seed variance are reported for Table 1. Since improvements are very close to 1% depending on the benchmark, error bars matter (especially for AIME with mean@32 sampling).
4. Only two unsupervised baselines are considered. R‑Zero reportedly “failed” to train on OpenMath‑Nemotron‑1.5B, but the paper does not detail alignment of budgets or hyper‑params or the failure mode. Including additional self‑play or co‑optimization baselines (e.g., URPO‑style, self‑rewarding corrections) and equalizing compute would make the comparisons more robust.

**Questions:**

Suggestions:
1. Maybe providing error bars for Table 1 tasks that this paper's method is **not significantly outperformed** by other baselines will be more convincing.
2. Making figure captions larger will be better for reading. For example, the text showing models and tasks in Figures 3 & 4 is small, and the author can consider making them as titles for each figure, not as captions.
3. Quantify the Coach warm‑up corpus and show full results with no warm‑up for all base models. Clarify the marketing claim: "data‑free Player training with optionally warmed‑up Coach," or similar.

Questions:
- For majority-vote pseudo-labels, as majority vote can reinforce systematic biases if the Player is consistently wrong. Did you explore additional verifiers (symbolic solvers, consistency under paraphrasing)?
- Why that exact range for difficulty filtering (0.2 to 0.8)?

---

> ### Author Response · Authors · 2025-12-01
>
> **Weaknesses 1 and Suggestions 3**:
>
> One point we want to make is that the warm-up of the coach model is not within the scope of our method. We used 4k PRIME mathematical data to warm up the coach, mainly to enable the coach to ask more constructive questions, thus facilitating co-evolution with the player. For R-zero implementations, their challenger also generates 3000-6000 problems for a cold start, freezing the solver's training. Here, "no external training data" means that no external data is involved when the coach and player are updating simultaneously. Our primary motivation is to help the player reach a new level of ability through the coach's guidance.
>
> We conducted an additional experiment on Qwen2.5-Math-7B-Instruct to show that the models can still co-evolve with guidance from Coach that have no warm-up.
>
> | | AMC | AIME2024 | AIME2025 | Minerva | MATH | Olympiad | OOD avg | avg |
> | :--- | :--- | :--- | :--- | :--- | :--- | :--- | :--- | :--- |
> | **Qwen2.5-Math-7B-Instruct**
> | BaseModel | 49.2 | 9.0 | 6.3 | 34.6 | 78.0 | 37.4 | 33.0 | 35.8 |
> | w/o Coach warm-up | 49.2 | 11.3 | 9.9 | 35.3 | 79.8 | 40.1 | **35.3** | **37.6** |
>
> ---
> **Weaknesses 2:**
>
> Thank you for highlighting the credit-assignment concern. We clarify below that each instruction is credited individually and instruction-level REINFORCE can avoid “good” instruction masking others.
>
> The Coach is not updated with a batch-level GRPO objective. Instead we use **instruction-level REINFORCE**:
>
> $$
> \nabla_\theta J(\theta)=\frac{1}{m}\sum_{i=1}^m R_i^{\text{Coach}}\nabla_\theta\log\pi_\theta^{\text{C}}(x_i),
> \qquad
> R_i^{\text{Coach}}=R_i^{\text{Player}}\cdot\Delta_t
> $$
>
> where
> - $R_i^{\text{Player}}=\frac{1}{n}\sum_{j=1}^n r_{i,j}$ is local to instruction $x_i$  (how well the Player solved this problem),
> - $\Delta_t$  is global for the whole batch (did the Player’s validation accuracy improve?),
> - but **each gradient term is weighted by its own $R_i^{\text{Coach}}$**.
>
> Hence a single “lucky” instruction cannot mask the others, as good instructions receive high individual weights and poor ones receive low weights. This multiplicative reward embodies a training principle: proposed tasks receive positive reinforcement only when they simultaneously achieve high Player performance during training (high $R_i^{\text{Player}}$) and contribute to measurable learning progress (positive $\Delta_t$).
>
> ---
> **Weaknesses 3 and Suggestions 1**:
>
> Thank you for your detailed review and valuable suggestion. We report error bars in Table 1 to reflect decoding uncertainty (independent rollouts per method). We therefore expect the reported improvements to be robust against decoding stochasticity. Full results will be included in the revised paper correspondingly.

---

> ### Author Response · Authors · 2025-12-01
>
> **Weaknesses 4** :
>
> Thank you for the insightful suggestions. We agree that broader baselines and tighter experimental alignment strengthen the empirical picture. Below we clarify (1) why the two unsupervised methods we chose are already the most representative, (2) how the methods perform when we align data and compute for R-Zero and CPMöbius, and (3) the exact failure mode we observed on OpenMath-Nemotron-1.5B.
>
> **(1) Choice of unsupervised baselines**
>
> RENT is one of the current state-of-the-art unsupervised RLVR method: it outperforms TTRL on the same mathematical benchmarks. We therefore treat it as the canonical baseline for unsupervised RLVR training.
> R-Zero, which initializes Challenger and Solver of the same model that interact adversarially, is the closest self-play analogue to our Coach–Player loop: one role generates tasks, the other attempts to solve them, and no human labels are used. Its adversarial design makes it the natural reference for collaborative vs. competitive dynamics.
>
> **(2) Alignment of budgets, hyper-parameters and compute**
>
> 2.1 Alignment of data budget (cold-start)
>
> - CPMöbius: 4 k PRIME problems used to warm-up the Coach only (solver frozen).
> - R-Zero: authors generate 3–6 k problems to warm-up the Challenger (solver frozen).
> Hence both pipelines consume an identical level of seed corpus for the proposer warm-up.
>
> 2.2 Alignment of hyper-parameters and compute budget
>
> We add a new experiment that align the hyper-parameters and compute budget between R-Zero and CPMöbius. The training process of R-Zero involves alternating phases: first training a 5-step questioner, followed by a 15-step solver, repeated three times, resulting in a total of 60 steps, solver global batch size: 128, number of rollouts: 5, challenger global batch size: 128, number of rollouts: 4. Thus, we utilized the checkpoint from our 60th step, which the parameter is: both coach and solver train batch size: 16, both coach and solver number of rollouts: 16, and compared it with R-Zero’s final training outcomes. The context length of all models remains consistent. The results are as follows:
>
> | | AMC | AIME2024 | AIME2025 | Minerva | MATH | Olympiad | OOD avg | avg |
> | :--- | :--- | :--- | :--- | :--- | :--- | :--- | :--- | :--- |
> | **Qwen2.5-Math-1.5B** |
> | R-zero | 39.2 | 9.8 | 5.0 | 19.3 | 62.4 | 26.8 | **24.7** | **27.1** |
> | CPMöbius | 40.0 | 10.3 | 4.9 | 17.3 | 59.3 | 29.3 | 24.2 | 26.9 |
> | **OctoThinker-3B-Hybrid-Zero**|
> | R-zero | 25.9 | 2.0 | 0.3 | 14.6 | 58.1 | 22.3 | 19.5 | 20.5 |
> | CPMöbius | 24.7 | 7.3 | 1.0 | 22.4 | 58.1 | 26.1 | **23.0** | **23.3** |
> | **Qwen2.5-Math-7B-Instruct**
> | R-zero | 50.5 | 9.5 | 7.4 | 32.7 | 83.3 | 38.1 | 34.2 | 36.9 |
> | CPMöbius | 49.0 | 11.3 | 9.6 | 34.9 | 79.5 | 38.2 | **34.7** | **37.1** |
>
> As shown, our method slightly underperforms R-Zero on the Qwen2.5-Math-1.5B model but achieves stronger performance on the other two models. Moreover, our approach can further enhance model capabilities. According to Section 5.4 of the R-Zero paper, its performance converges after three iterations. Therefore, we believe that under comparable computational budgets, our method still holds an advantage. And the improvement of model capability stems from our framework raising the performance upper bound, rather than simply leveraging additional compute.
>
> **(3) Failure mode of R-Zero on OpenMath-Nemotron-1.5B**
>
> When training Openmath-Nemotron-1.5B, R-Zero aims to train the model to generate questions wrapped within \<question\>\</question\> tags, which are then extracted for solver training. However, during our reproduction, we observed that the trained Openmath-Nemotron-1.5B model frequently failed to comply with the required formatting. Specifically, when prompted to "generate a question wrapped in \<question\> tags":
>
> - The model often did not wrap questions in the specified \<question\>\</question\> tags for ~ 90% of the time. Instead, it used various alternative free-form formats such as randomly starting with "Problem Setup:", "Consider the problem:", "Problem:" or "Problem Statement:", which are hidden between \<think\>\</think\> tags and could not be properly extracted by the R-Zero pipeline.
> - For ~ 10% of the time, the model generated a significant amount of repetitive and meaningless content, or occasionally reproduced the prompt itself verbatim, such as outputting strings like:
> \<question\>\n{The full problem statement on one or more lines} [Problem ]\n\</question\>\n\n
>
> Because R-Zero’s pipeline relies on extraction of the \<question\> tag to build the solver training set, no usable questions can be obtained after the first iteration, causing the training pipeline to fail.
> To solve this problem, our Coach warm-up explicitly finetunes the model to ask questions in a consistent format, and that's why CPMöbius does not suffer from this formatting issue.
> We appreciate the suggestion and will include a more detailed discussion of this issue in the revised version of our paper.

---

> ### Author Response · Authors · 2025-12-01
>
> **Questions 1** :
>
> The work of TTRL[1] demonstrates that majority voting, a test-time scaling method, is a very efficient and suitable way to promote RL training and reward shaping. In our work, majority voting is used as a method that is simple and without verifier supervision and CPMöbius is a general framework, which only requires an efficient blackbox verifier:
> $$
> f(x,y_{1:k}) \to r \in \mathbb{R}
> $$
> which induces a reward score from the problem xxx and candidate solutions $y_{1:k}$. majority voting is only one concrete instantiation of $f$; other choices such as symbolic solvers, reward models, interpretive consistency, etc., are fully compatible with CPMöbius.
> We choose majority voting instantiation of $f$ since it has already had robust improvements. Therefore, We did not, in this submission, perform a huge amount of experiments with full symbolic solvers or consistency under paraphrasing verifiers, mainly due to engineering and compute constraints.We agree these are promising directions to mitigate the failure mode mentioned by the reviewer. We will explicitly discuss symbolic solvers and consistency under paraphrasing verifiers checks as natural, stronger instantiations of $f$ that we plan to explore in follow-up work.
>
> [1] Zuo, Yuxin, et al. "Ttrl: Test-time reinforcement learning." arXiv preprint arXiv:2504.16084 (2025).

---

> ### Author Response · Authors · 2025-12-01
>
> **Questions 2**:
>
> The 0.2-to-0.8 range is used because it keeps the "just-right" problems: not so easy that the model learns nothing new, and not so hard that it gets only garbage feedback.
>
> The same "moderate-difficulty" rule is used in other RL work: keep items whose pass-rate is between 20 % and 80 %; everything outside this window is either un-informative or too noisy [1]. Several works also introduce their data preparation pipeline to construct challenging datasets for RL training, providing inspirations on difficulty rating. Light-R1[2] and Skywork-OR1[3] conduct offline data selection that leverages a trained checkpoint to sample and verify responses for the query of each training sample, keeping only the samples with a moderate pass rate, and filtering out the samples with overly high or low pass rates, indicating that the corresponding queries are either too easy or too hard. DeepScaleR[4] revealed that samples with an overly high pass rate are too easy for model training, while samples with a zero pass rate are often unverifiable or contain errors, therefore, both should be filtered out. Open-Reasoner-Zero[5] further adopts this filtering strategy to construct training data from synthetic and distilled data that is more noisy.
>
> Empirically, the band was the best-balanced among the thresholds that we tested. We assume the range for difficulty filtering is $(\alpha, 1 - \alpha)$. Across $\alpha \in {0, 0.1, 0.2}$, where (TLow, THigh) pairs are {(0,1), (0.1,0.9), (0.2,0.8)}, the setting (0.2,0.8) gave the highest average score on the test sets while still retaining a reasonable number of training samples. Note that in Table 2, the "w/o Instruction Filter" ablation study results shows that disabling this mechanism reduces average accuracy from 28.8% to 24.9% (OOD: from 26.8% to 22.5%), confirming that appropriate difficulty calibration is crucial for efficient learning. When $\alpha \geq 0.3$, the condition become too difficult for the Player to generate consistent-enough answers for the Coach's instructions, and the training efficency will be low due to filtering too many instructions.
>
> [1] Bae, Sanghwan, et al. "Online difficulty filtering for reasoning oriented reinforcement learning." arXiv preprint arXiv:2504.03380 (2025).
>
> [2] Liang Wen, et al. "Light-r1: Curriculum sft, dpo and rl for long cot from scratch and beyond." arXiv preprint arXiv:2503.10460, 2025a.
>
> [3] Jujie He, et al. "Skywork open reasoner series."  https://capricious-hydrogen-41c.notion.site/Skywork-Open-Reaonser-Series-1d0bc9ae823a80459b46c149e4f51680 , 2025a. Notion Blog.
>
> [4] Michael Luo, et al. "Deepscaler: Surpassing o1-preview with a 1.5b model by scaling rl." https://pretty-radio-b75.notion.site/DeepScaleR-Surpassing-O1-Preview-with-a-1-5B-Model-by-Scaling-RL-19681902c1468005bed8ca303013a4e2 , 2025b. Notion Blog.
>
> [5] Jingcheng Hu, et al. "Open reasoner-zero: An open source approach to scaling reinforcement learning on the base model." https://github.com/Open-Reasoner-Zero/Open-Reasoner-Zero, 2025.

---

### Meta-Review · Area_Chair_aX1K · 2025-12-28

**Summary:**

Reviewer zyLm, 7o3P both critized the "data‑free" training setting, specifically, the Coach is not built from scratch but initialized from Qwen2.5-7B-Instruct and then cold-started with PRIME Eurus-2-RL-Data. Thus reviewers feel such a claim maybe oversold and should be either tuned down or revised with more clear position.

Reviewer zyLm, 7o3P, and cag2 raised concerns about the significance of improvement compared to existing baselines. Reviewer zyLm mentioned missing error bars or variance with 1% improvement. Reviewers 7o3P have concerns on fair comparison with RENT and suggested to add more details on training stability. Reviewer cag2 critized that the performance of the proposed method is noticeably lower than its non-data-free counterparts.

Reviewer Ewjr, 7o3P raised concerns about the overfitting risk and generalization capability of obtaining feedback for the Coach based on AMC benchmark.

Reviewer 7o3P and cag2 both concerns about the OOD generalization performance/setting in this manuscript.

Other minor concerns including writting, visualization, training and evaluation details are not informed AC's final decision making.

**Reviewer Concerns:**

The authors try to address reviewers' concern on the dependency of instruction-tuned Coach model and warm-up dataset by aligning the experimental setting with previous work, i.e., R-zero, providing additional results without warm-up. AC thinks the response make sense and the reviewers' concern on this point can be addressed.

Regarding the improvements over counterparts and OOD generalization setting/capabilities, these concerns may be still outstanding for reviewers, strong arguments may still be necessary to address these concerns.

**Reviewer Scores:**

Reviewer zyLm and EwJr are expected to maintain their original rating after discussion. Reviewer cag2 and 7o3P's major concerns are still outstanding and AC is not sure whether they will change their original rating.

The authors made great efforts in addressing reviewer's concerns. AC appreciates the heavy workload managed by the authors during the review period and acknowledges that these results strengthen the overall quality of the paper. From AC's own reading of this manuscript, the proposed training framework is novel and promising, which distrinct from existing RL methods.

However, given the substantial volume of new experimental results and the fact that 1) the required sum-total modification are not minor and 2) some major concerns seem still exist after a few discussion, this manuscipts is not ready to be published in its current revision and further review are needed in the future. The authors are encouraged to incoparate the reviewers' suggestions, e.g., clearifying or narrowing research position, and providing more compelling experimental results in their revisions.

---

### Decision · Program_Chairs · 2026-01-26

Reject